

# On the potential of a neural network-based approach for estimating XCO2 from OCO-2 measurements

François-Marie Bréon, Leslie David, Pierre Chatelanaz, Frédéric Chevallier

Laboratoire des Sciences du Climat et de l'Environnement/IPSL,
CEA-CNRS-UVSQ, Université Paris-Saclay, F-91191 Gif-sur-Yvette, France

*Correspondence to*: Francois-Marie Breon (fmbreon@cea.fr)

**Abstract**. In David et al (2021), we introduced a neural network (NN) approach for estimating the column-averaged dry air mole fraction of $CO_2$ (XCO2) and the surface pressure from the reflected solar spectra acquired by the OCO-2 instrument. The results indicated great potential for the technique as the comparison against both model estimates and independent TCCON measurements showed an accuracy and precision similar or better than that of the operational ACOS (NASA's
Atmospheric $CO_2$ Observations from Space retrievals – ACOS) algorithm. Yet, subsequent analysis showed that the neural network estimate often mimics the training dataset and is unable to retrieve small scale features such as $CO_2$ plumes from industrial sites. Importantly, we found that, with the same inputs as those used to estimate XCO2 and surface pressure, the NN technique is able to estimate latitude and date with unexpected skill, i.e. with an error whose standard deviation is
only 7° and 61 days, respectively. The information about the date mainly comes from the weak $CO_2$ band, that is influenced by the well-mixed and increasing concentrations of $CO_2$ in the stratosphere. The availability of such information in the measured spectrum may therefore allow the NN to exploit it rather than the direct $CO_2$ imprint in the spectrum, to estimate XCO2. Thus, our first version of the NN performed well mostly because the XCO2 fields used for the training were
remarkably accurate, but it did not bring any added value.
Further to this analysis, we designed a second version of the NN, excluding the weak $CO_2$ band from the input. This new version has a different behaviour as it does retrieve XCO2 enhancements downwind of emission hotspots, i.e. a feature that is not in the training dataset. The comparison against the reference Total Carbon Column Observing Network (TCCON) and the surface-air-
sample-driven inversion of the Copernicus Atmosphere Monitoring Service (CAMS) remains very good, as in the first version of the NN. In addition, the difference with the CAMS model (also called *innovation* in a data assimilation context) for NASA Atmospheric $CO_2$ Observations from Space (ACOS) and the NN estimates are significantly correlated.
These results confirm the potential of the NN approach for an operational processing of satellite
observations aiming at the monitoring of $CO_2$ concentrations and fluxes.

## 1. Introduction

There is a growing interest for the monitoring of $CO_2$ from space. The aim is not so much the atmospheric concentration, which is already known with high accuracy, but rather the $CO_2$ fluxes. Indeed, there is a need to monitor natural fluxes of $CO_2$ to better understand their driving factors
and to improve land and ocean models. There is also a strong societal requirement to monitor the $CO_2$ anthropogenic emissions at national and more detailed scales. For these objectives, a series of dedicated instruments have been put in orbit since the Greenhouse Gases Observing Satellite (GOSAT, Yokota et al., 2009) and the second Orbiting Carbon Observatory (OCO-2 Eldering et al., 2017), launched in 2009 and 2014, respectively, and still operated at the time of writing. This new



and evolving constellation is directly supported by Japanese, US, Chinese and European space agencies (CEOS Atmospheric Composition Virtual Constellation Greenhouse Gas Team, 2018). The OCO-3 instrument was launched in 2019 and is flying attached to the International Space Station (ISS) with a focus on the imagery of cities and industrial sites (Taylor et al., 2020). These targets are also the main focus of the CO2M mission under development at ESA.

These missions all use the same general principal to estimate the $CO_2$ concentration in the atmosphere. They measure the reflected solar light at high spectral resolution, which allows identifying absorption lines whose depth is related to the total amount of gas along the atmospheric path. Atmospheric $CO_2$ shows a number of such lines close to 1.61 and 2.06 µm so that these spectral regions are targeted. Because the absorption is more intense at 2.06 µm, this measurement

channel is often referred to as the *strong-CO2* (or sCO2) band, whereas the 1.61 µm is the *weak-CO2* (wCO2) band. The line depth is also affected by the surface pressure and the amount of scattering particles in the atmosphere. To identify and account for their contribution, an additional measurement is made around the oxygen absorption band at 0.76 µm ($O_2$ band). The combination of these measurements makes it possible to estimate the column-averaged dry air mole fraction of

$CO_2$, referred to as XCO2 (Crisp et al., 2004). Note that the MicroCarb instrument, to be launched by CNES in 2022, will have a fourth band at 1.27 µm. This band serves the same purpose as the $O_2$ band; it has the advantage of being spectrally closer to the CO2 bands and the disadvantage of being affected by airglow (Bertaux et al., 2020).

The interpretation of measured spectra in terms of XCO2 is achieved through *full physics* algorithms that explicitly account for the absorption by $CO_2$, $O_2$ and water vapor, for scattering in the atmosphere and for non-lambertian reflection on the Earth surface. The modeling must also account for the instrument line shape function and doppler effects. The inversion process is iterative and starts from a prior estimate of all atmospheric parameters. It is very computer-time

consuming. The processing of OCO-2 data has shown systematic differences between the measured spectra and those modeled after inversion which led to the development of empirical corrections to the measured spectra (Crisp et al., 2012; O'Dell et al., 2018). In addition, raw XCO2 retrievals show significant biases against reference ground-based retrievals (Wunch et al., 2011b, 2017). These biases, together with the comparison against modelling results, led to the

development of empirical corrections to the retrieved XCO2.
The need for empirical corrections to the full-physics algorithms and the considerable computer load motivated us to develop an alternative approach described in David et al. (2021). We used an artificial network technique (NN) which is purely empirical, without the use of any radiative transfer model. Our hypothesis was that the CAMS (Copernicus Atmosphere Monitoring Service,

https://atmosphere.copernicus.eu/) model constrained by surface air-sample measurements provides a fairly accurate estimate of the atmospheric $CO_2$ concentration, including the growth rate over multiple years. Indeed, the seasonal cycle of $CO_2$ together with the growth rate generate a set of XCO2 samples with a well-known variability. The uncertainties on the modeling are small with respect to the range of XCO2 samples that is available in the multi-year dataset. As a consequence,

although CAMS is not the truth, it may be used for supervised learning.
In practice, we used a series of OCO-2 spectra from a 5 year-dataset for the NN training. We then applied the NN to the observations that were not used in the training and compared their estimates to both the same CAMS model used for the training and also the fully-independent set of Total Carbon Column Observing Network (TCCON, Wunch et al., 2011a) observations. The results

indicated an accuracy and precision that were similar, if not better, to that of the ACOS algorithm. More recent results challenged our interpretation of the NN skill. In particular, the XCO2 estimates of the NN did not show significant enhancement downwind of large power plants, unlike the product of the NASA Atmospheric $CO_2$ Observations from Space (ACOS) full-physics algorithm. This is shown in the following together with our interpretation. A new version of the NN resulted

from this interpretation, that retains the high accuracy of the first version, while being much more independent from the training dataset.



In the following, Section 2 describes the main characteristic of the NN approach and the training procedure. Section 3 presents the limitation of the first version of the NN, as it shows no innovation with respect to the training dataset. Section 4 describes and justifies a new version of the NN approach. Section 5 discusses the results, suggests directions for improvements, and concludes.

## 2. Data and method

The NN described in this paper estimates XCO2 from spectra measured by the OCO-2 satellite over land. Most of the analysis is made with the spectra acquired in nadir mode, but we have also developed a version for glint acquisition that is described and commented at the end of section 4. Conversely to the analysis in David et al. (2021) we now use all cross-track footprints.

We use spectral samples in the three bands of the instrument (around 0.76, 1.61 and 2.06 μm). They have footprints of ~ 3 km$^2$ on the ground. In principle, each band is described by 1016 samples but some are marked as bad either because some of the corresponding detectors died at some stage or because of known temporary or permanent issues. We systematically remove 15 spectral samples that are flagged in about 80% of the spectra and 478 pixels in the band edges. Conversely, we do not remove the samples that are affected by the deep solar lines, and we let the NN handle these specific features. Because the information in the spectrum is mostly in the relative depth of the absorption lines, and not in their overall amplitude, we normalize each spectrum by a radiance that is representative of the offline values (i.e. the mean of the 90-95% range for each spectrum). This essentially removes the impact of the variations in the surface albedo and in the solar irradiance linked to the sun zenith angle.

Figure 1 offers a graphical representation of the NN. As input, we use the three band spectra (or a subset, see below), the observation geometry (Sun and view zenith angle: SZA and VZA, and relative Azimuth: AZI). Some versions also use the surface pressure (Psurf) as input. No explicit information is provided to the NN regarding the location or date of the observation. The inputs feed all the neurons of a first "hidden" layer. We use a fully connected neural network, which means that all the neurons are connected to the neurons of the previous and next layer. We have attempted NN versions with a variable number of hidden layers (a single one was used in David et al. (2021)). Each neuron computes a weighted sum of the inputs and derives a single output on the basis of either a sigmoid function or a "rectified linear unit". The weights of the input variables to the neurons are adjusted iteratively with the standard Keras library (Keras Team, 2015) for an optimal agreement between the NN output and a reference.

The NN training is based on OCO-2 radiance measurements (v10r) acquired between February 2015 and December 2019. We make use of XCO2 estimates and the quality control filters of the ACOS L2Lite v9r products: only observations with *xco2_quality_flag*=0 are used. For the validation of the NN estimates, we also use observations with relaxed quality requirements. For versions of the NN that use the surface pressure as input, we use the estimate that is provided together with the OCO-2 data and that is derived from the Goddard Earth Observing System, Version 5, Forward Processing for Instrument Teams (GEOS5-FP-IT) created at Goddard Space Flight Center Global Modeling and Assimilation Office (Suarez et al. 2008 and Lucchesi et al. 2013). The weather model pressures have been adjusted to the sounding surface height.

Our analysis makes use of the CAMS CO₂ atmospheric inversion (Chevallier et al., 2010; https://atmosphere.copernicus.eu/, version 19r1). This product was released in July 2020 and contributed, e.g., to the Global Carbon Budget 2020 (Friedlingstein et al., 2020). It results from the assimilation of CO₂ surface air-sample measurements in a global atmospheric transport model run at spatial resolution 1.90° in latitude and 3.75° in longitude over the period 1979-2019 and using the adjoint of this transport model. Neither satellite retrievals nor TCCON observations were used for this modelling. For each OCO-2 observation, XCO2 is computed from the collocated concentration vertical profile, through a simple integration weighted by the pressure width of the model layers.



Note that the model layers use "dry" pressure coordinates so that there is no need for a water vapor correction in the vertical integration. The $XCO_2$ from CAMS is used both for the training and the evaluation, although using independent datasets: The "training" dataset is a 3% random sample of the full dataset. The observations that are used for the training are earmarked and not used for further evaluation.

### 3.  Initial results and interpretation

David et al (2021) described a first version of the NN approach to estimate $XCO_2$. In this first version, the surface pressure was not used as input, the training was made on observations acquired during even months, while the validation used observations of the odd months. The results were surprisingly good as the statistical difference to both the CAMS modeling and the independent TCCON observations indicated an accuracy similar or better than that of the CAMS product. Further analysis posterior to the publication were worrisome however.

First, we found that well documented local enhancement of $XCO_2$ in the ACOS product (e.g., Nassar et al., 2017; Reuter et al., 2019), also referred to as *plume*, did not show up in the NN product. We analyzed in particular a case over South African acquired on August, 31st, 2016 an illustration of which is provided on Figure A1. Over a distance of ≈100 km, the ACOS product shows several well identified enhancements of ≈5 ppm, whereas the NN product does not show any significant pattern. The presence of large coal power plants upwind of the OCO-2 observations makes the enhancements trustworthy. We found many similar cases where the NN did not display an $XCO_2$ plume where ACOS did. We concluded that the NN did reproduce the seasonal variation of $XCO_2$ together with the growth rate but was unable to identify small-scale features. Since all observations are processed independently, we could not interpret this apparent incoherence.

Second, we made an experiment where the training dataset is biased by 1 ppm for the observations acquired during a single month (within the full period of 50+ months). When applied to the validation dataset, the differences to CAMS show a bias of ≈0.5 ppm but only for the observations that are within a few weeks of the biased period. This is rather surprising as the observation date is not an input of the NN. Still, these results provide a clear indication that this version of the NN is somehow sensitive to the observation date.

To investigate the issue, we developed and trained a new NN with the same inputs, but aiming at estimating the date, latitude and longitude. For the training, we used the true values of these parameters and we analyzed how the NN was able to make an estimate based on the inputs (the spectra and the observation geometry). Figure 2 shows the histograms of the errors when applied to the independent dataset.

The results indicate that the NN approach is able to make a reasonable estimate of the location and date of the observation based on the spectra and the observation geometry. The standard deviation of the latitude estimate is on the order of 7°. One may expect that this information is largely derived from the observation geometry that changes with the latitude (both the SZA and the azimuth do). One argument in favor of this hypothesis is that the longitude estimate is much worse, with a standard deviation on the order of 58°. Indeed, for a given day, the observation geometry is nearly the same for all successive orbits; thus, there is no information in the observation geometry to estimate the longitude, while there is such information for the latitude. As for the date, the standard deviation is ≈ 61 days, or 2 months. Clearly then, in the input data of the NN, there is indirect information about the observation date and latitude and this was a surprise to us. Indeed, when describing the NN approach in David et al (2021), we argued that the NN had no information on the measurement date, as successive observations from the same day-of-year and location, but different years, were made with the exact same observation geometry.

The various histograms of Figure 1 were made using a single ($O_2$) band, a combination of $O_2$ band with either $CO_2$ bands, and all three bands. The most striking difference between the various histograms is for the date estimate. Indeed, the accuracy strongly degrades when the $wCO_2$ band is not included. The combination of $O_2+wCO_2$ bands leads to a much better accuracy (a factor of



more than 3 on the standard deviation) than that obtained with O2+sCO2. The other differences on the histograms are not as large.

How does the NN gets an indirect information on the observation date, and why is this information somehow contained in the wCO2 band? Our best interpretation is that the weak $CO_2$ spectrum is sensitive to the upper atmosphere $CO_2$ concentration that is rather well mixed while increasing regularly in time. The absorption lines in the sCO2 band are much stronger so that their centers are saturated in the spectra. As a consequence, the $CO_2$ signal is more in the line wings which are more sensitive to the higher pressure (lower altitude) levels. The wCO2 lines are not saturated and the spectrum shape may provide the information for an estimate of the high-altitude $CO_2$ concentration. We investigated another hypothesis that the wCO2 detector shows an evolution in time. However, we did not find any indication of such behavior. Thus, at this point, the stratospheric $CO_2$ hypothesis is physically plausible and is our best hypothesis because of no other. Note however that we have investigated the correlation between the longitudinal anomalies of stratospheric $CO_2$ in the CAMS model and the error on the date estimate by the NN approach. No such correlation was found. Thus, either our hypothesis is wrong or the description of the longitudinal variations of stratospheric $CO_2$ in CAMS offer a poor representation of the reality. Both hypotheses are plausible.

These results clearly demonstrate that the input data to the NN provides an indirect information on the date and latitude. Atmospheric simulations such as those of CAMS indicate that XCO2 variations are mostly a function of time and latitude. Indeed, the typical deviations of XCO2 along the longitudes are on the order of 0.5 ppm (standard deviation). Thus, we hypothesize that our first version of the NN, as published in David et al. (2021) obtains a proxy of the latitude and date, and outputs the corresponding CAMS value. Based on the CAMS simulation, we found that the typical uncertainty on the position and date ($\sigma_{lat}=7°$, $\sigma_{lon}=58°$ and $\sigma_{date}= 60$ days) leads to a 1-sigma error of 0.91 ppm on XCO2 (difference between the values at the true and perturbated location/date). This value appears consistent with the precision obtained with our first version of the NN.

## 4. A new version of the neural network

As shown above, the NN appears to use the wCO2 band to derive a proxy of the observation date which makes it possible, together with the proxies of the location, to estimate XCO2 based on the statistical distribution of the CAMS XCO2. To avoid this feature, an option is to not use the information from the wCO2 band. We therefore developed a similar version of the NN but without this band (i.e. only the O2 and sCO2, together with the observation geometry). With this version, the behavior of the NN changes markedly. The most important feature is that the NN now reproduces the XCO2 plumes that are shown by the output of the ACOS algorithm. Two representative examples are shown in Figure 3. These cases demonstrate that the NN does produce XCO2 features that are not in the training database, as we expected. The NN is trained on the variations of XCO2 caused by the atmospheric growth rate and the surface flux seasonal cycle. It identifies signatures in the spectra that relate to the $CO_2$ atmospheric content. These signatures can then be used for an estimate of XCO2, even for situations that are poorly reproduced in the training dataset.

In addition to the change in the band selection, and posterior to the result shown on Figure 3, we made several other modifications to the NN algorithm:

First, we decided to use the surface pressure from the weather forecast model as an additional input to the NN. In David et al. (2021), the surface pressure was an output of the NN model. It was used to demonstrate the capability of the NN approach to interpret the spectral shapes in terms of atmospheric parameters. Indeed, the estimate of the surface pressure could be compared to an independent estimate from numerical weather analyses which are known to be precise within ≈1 ‰. However, the surface pressure may alternatively provide useful information to the NN for the interpretation of the spectra, as it does in full-physics algorithms in the form of a prior estimate.



Second, we decided to increase the number of NN hidden layers to 5 (instead of 1 in David et al. 2021). Our experience indicates that, with a larger number of layers, there is less over-fitting of the training spectra, i.e. there is a better agreement between the loss of the training and that of the test dataset. An increased number of hidden layers also leads to a slightly better performance, in particular for the NN that was designed for the land-glint observations (see below).

Third, we developed a similar approach for the glint cases (still over land). Our initial fear was that it would be more difficult for the NN to handle glint observations because of (i) larger variations in the optical path than for the nadir mode and (ii) the doppler effect that may affect the absorption line positions on the input spectra. This is why our first attempts focused on the nadir cases, but there is a need to also exploit the many observations acquired in glint mode.

Figure 4 shows the inter-comparison of the XCO2 estimated from CAMS, ACOS and NN. All three datasets are highly consistent, with a statistical difference around 1 ppm and little bias. Let us recall that there is no satellite data input to CAMS so that it is fully independent from ACOS. The 1.06 ppm standard deviation of their differences demonstrates that both product precisions are better than this number. CAMS and the NN are not as independent because the latter is trained with
the former (but using different space-time locations). Still, it has been shown that the NN retrieves features that are not in CAMS which indicates some independence between the satellite product and the model. The standard deviation of their differences is 0.85 ppm. The quadratic difference between NN and ACOS is a strong function of the sCO2 albedo as shown in Figure 5: it decreases from ≈1.5 to ≈0.75 ppm as the O2 band albedo increases from 0.10 to 0.45. A better accuracy of
the satellite product with stronger surface albedo is expected as (i) the measurement signal to noise gets higher and (ii) the relative contribution of atmospheric scattering to the signal decreases. Figure 4 also shows that the slope of the best fit of the satellite products against CAMS are small but of opposite side with respect to 1. As a consequence, there is a more significant slope deviation from 1 (0.97) between the two satellite products.

Figure 6 provides further information on the differences between the remotely-sensed products and the CAMS estimate. The histograms are close to Gaussian and confirm that NN is closer to CAMS than the ACOS counterpart. An interesting feature is that both the NN-CAMS and ACOS-CAMS differences depend on the cloud flag (*cloud_flag_idp*), which indicates that this flag has some value. The difference between the cloud contamination histogram remains small however, and does
not deserve to disqualify the observations with a cloud flag of 2. Here, we only use the "definitely clear" and "probably clear" cases (flags of 3 and 2). The population of the lower value cases ("definitely cloudy" and "probably cloudy") is much smaller and the histogram for these cases are not shown, while they show further degradation. It is difficult to elaborate further as the true nature of the cloud contamination in the cases classified as "probably clear" is unknown.


Figure 7 is based on the satellite product innovation, i.e. the difference to the model estimates. Indeed, one may consider that the model provides current knowledge on the XCO2 distribution, constrained by surface air-sample measurements and atmospheric transport. The satellite product has the potential to improve this knowledge, but only as much as the difference with the model
estimate. Typical values are around 1 ppm. The interesting result brought by Figure 7 is that the two satellite estimates are significantly correlated. This provides further evidence that the NN estimate is not only a reconstruction of the training dataset (CAMS) with some noise. Indeed, when NN differs from the model, ACOS, the independent satellite product, tends to agree.

Finally, Figure 8 shows a comparison of the model and remotely sensed estimates of XCO2 against the reference retrievals of the TCCON network. Although the OCO-2 satellite platform can be oriented so that the instrument field of view is close to the surface station, we only use here nadir data. Indeed, the NN was not trained on the target data and can only be used to process measurements that have been acquired in observation configurations that are similar to those of the
training. We thus have to rely on nadir or glint measurements acquired in the vicinity of TCCON



sites. In the following, we use nadir measurements that are within 5 degrees in longitude and 1.5 degrees in latitude to the TCCON site. For the reference, we average the TCCON estimates of XCO2 within 30 minutes of the satellite overpass. No attempt was made to correct for the different weighting functions of the surface and spaceborne remote sensing estimates. Statistics per station

are provided in Table 1. The biases vary significantly among stations, although they are generally less than 1 ppm (in amplitude). Two stations, Pasadena and Zugspitze, show a large negative bias for both satellite estimates as well as the model. For Pasadena, it may be interpreted as the impact of the city on the atmosphere sampled by the TCCON measurement, while the atmosphere at the location of the satellite observation (which may be several hundred km away) is less affected.

Zugspitze is a high-altitude site (2960 m), so that the atmospheric column sampled by the sunphotometer does not have the same vertical representativeness as that of the satellite observation (in addition to the spatial distance, that is common with other sites). A large negative bias is also found at Eureka (80.05°N). The fact that the difference with the CAMS model at this site is much larger than for other sites could hint at an issue in the sunphotometer product there. Conversely,

there are large positive biases at Burgos and Ny-Ålesund (78.9°N, very close to the latitude of Eureka). Since the model and satellite estimates somewhat agree, one may also question the TCCON calibration at these sites. For other stations, which forms the large majority, the biases are smaller than 1 ppm and there is a fair consistency between the satellite products in the sense that the sign of their bias is the same in most cases. The range of the difference with TCCON varies among

stations. The best satellite-TCCON agreement is found at the Lamont station which, interestingly, is also the one with the most coincidences. Excellent agreements are also seen at Darwin, Edwards, Park Falls and Bremen. The comparison with TCCON does not allow favoring one satellite estimate versus the other. Focusing on the stations with a large number of observation (25 overpasses or more), the NN estimates appears slightly better than ACOS at Darwin, Edwards,

Garmisch, Orléans and Bialystok, while it is the opposite at Saga, Park Falls, and Sodankylä. The figure (and table) also clearly shows that the CAMS product offers a better agreement with the TCCON data than any of the satellite estimates in most cases, which questions the added value of the remote-sensing data at large scale and in delayed mode (about nine months after the date for the surface air-sample-driven CAMS product).


We have applied a very similar procedure to the OCO-2 observations acquired in glint mode over land. An evaluation of the estimate performance is shown in Figure 6, 7, A3 and A4. The conclusions are very similar to those obtained for nadir. The agreement with CAMS is slightly degraded with respect to the nadir cases (0.92 ppm vs 0.85 for the "certainly clear" observations)

but remains significantly better than that of ACOS (Fig. 6 and S3). The deviations from the model of the two satellite estimates are significantly correlated, and the correlation coefficient is even larger than that derived for nadir observations (0.45 vs 0.39, Figure 7). The comparison with the TCCON estimates leads to the same conclusions as those described above for the nadir cases.

## 5. Discussion and Conclusion

This paper follows on from David et al. (2021) in which we described a neural network-based technique to estimate XCO2 and the surface pressure from the OCO-2 spectral measurements. An important message is that our interpretation of the results in that earlier study was incorrect. The NN developed in that paper reproduced the statistical variations of the training dataset (CAMS) and was unable to generate features, such as plume from emission hot-spots. Thus, contrarily to our

claims, the NN method, as presented in that paper, could not be used to process OCO-2 and generate XCO2 estimates with any real value. We have shown here that a NN-based procedure is able to estimate the latitude and date of the observation with a reasonable accuracy. This was unexpected to us as we wrote in David et al. (2021) "Let us recall that the NN input does not contain any information on the location or date of the observation. This is a strong indication that

the information is derived from the spectra as the NN does not "know" the CAMS value that corresponds to the observation location". Because most of XCO2 variations are a function of





latitude and date, this information is used by the NN to generate a reasonable estimate, i.e. one that mimics the main variations in the training dataset.

A question remains on the indirect information that is used by the NN to estimate the observation date. The fact that the precision on the date estimate is much better when using a combination of the O2+wCO2 rather than of the O2+sCO2 suggests that the information lies in the wCO2 band. Our best hypothesis is that the wCO2 spectra contains some information on the stratospheric $CO_2$ whose concentration is well-mixed while increasing regularly with time and implicitly contains, therefore, an information on the observation date. Further testing this hypothesis would require, for

instance, the identification of some anomaly in the stratospheric $CO_2$ (linked to a specific atmospheric circulation) that would show up as a significant error on the date estimate made by the NN.

Despite this setback we have continued our analysis on the potential of the NN to process the OCO-2 spectra. A strong motivation relied on the results obtained for the estimate of the surface

pressure. Indeed, David et al. (2021) showed that the NN could estimate the surface pressure with an accuracy on the order of 3 hPa. The spatial and temporal variations of the surface pressure are very much larger than this number, so that the NN estimate cannot rely on some indirect information on the date and location. This provides a strong indication that the NN method has the potential to extract meaningful information from the spectra.

We have therefore developed a new version of the NN excluding the wCO2 band from the inputs. In this version, the behavior of the NN is very much different from the earlier version as it generates features that are not in the training dataset. This clearly shows that the NN uses the signature of XCO2 contained in the sCO2 spectra to make an XCO2 estimate. The accuracy of this estimate is similar to the one obtained with the first version of the NN, and similar to that of the ACOS

products. This is confirmed by the comparison of the XCO2 estimates against the TCCON retrievals. Another strong argument that the NN XCO2 estimate contains true information and is not only a noisy copy of the training dataset is that the innovations of the two satellite estimates, i.e. the differences to the model data, are significantly correlated (Fig. 7).

These results confirm that the NN technique has a strong potential to process the OCO-2

observations, as well as those from forthcoming missions aiming at the observation of $CO_2$ from space such as the forthcoming MicroCarb (Pascal et al. 2017) or the CO2M constellation (Sierk et al. 2019). As discussed above, the current version does not use the wCO2 band at all and this may be seen as a loss of useful information. There is therefore a need to select appropriate spectral samples in the wCO2 band rather than discarding them all. It requires improved understanding of

the indirect information that is used by the NN to estimate the observation date and location.

The NN technique has two obvious advantages compared to the physical methods that are used to process the OCO-2 observations as well as other instruments with similar objectives: (i) a much smaller computational burden and (ii) no need for a de-bias procedure (O'Dell et al 2018, Kiel et al.

2019). Our implementation still faces remaining challenges, that we discussed in David et al. (2021).

The first challenge is the absence of a quality indicator. With the physical methods, the spectrum residuals provide an efficient mean to identify cases when no satisfactory agreement can be found between the measured and modelled spectra. Although the NN described here aims at an estimate

of XCO2, we have shown earlier that the same tool can be used for an estimate of the surface pressure with a 1 sigma precision on the order of 3 hPa for clear-sky cases. Numerical weather analyses are actually better than that. Thus, one may use the comparison of the surface pressure estimate from the NN to the numerical weather data for an easy identification of perturbations to the spectra that are linked to cloud or large aerosol contamination. This would allow an easy and rapid

quality indicator for the selection of observations that may be used for XCO2 estimates, either using a physics-based algorithm or a NN approach. To provide precision estimates for each NN-based XCO2 estimate, ensembles of randomized trainings, where uncertain parameters or input/output



variables are varied adequately (e.g., Chau et al. 2021), or analytical estimates (Aires et al., 2004) should be explored.

The second challenge concerns the absence of a quantitative indication of the amount of information that the NN takes from its prior information (contained in the training database) vs. the amount of information that the NN takes from the measured spectra. For Bayesian full-physics retrievals, these weights are represented by the *averaging kernel* (Rodgers, 1990) that allows a clean comparison of each retrieval with 3D atmospheric models, at least in theory (see the
discussion about the practical difficulties in Chevallier, 2015). The NN training targets the $CO_2$ column with a homogeneous weighting along the vertical but this can hardly be achieved without some contribution from the prior information. This challenge may be evaluated in the future on the basis of radiative transfer simulations.

The last challenge concerns the need for a high-quality training dataset. The comparison against the
TCCON observations (Figure 8) demonstrates that the CAMS inversion product meets this requirement. In fact, there are strong indications that CAMS remains better than the satellite products, at the very least in terms of global precision. However, because of the atmospheric growth rate of $CO_2$, the training must be regularly updated. Indeed, with a frozen training dataset, the true real-time XCO2 progressively leaves the training range. The NN approach requires a
training dataset that is representative of the observation and would then lead to underestimates. For quasi-near-real time data assimilation (e.g., Massart et al., 2016), the training dataset must therefore gradually integrate recent high-quality XCO2 data, but without sacrificing robustness.

## Acknowledgments

This work was in part funded by CNES, the French space agency, in the context of the preparation for the MicroCarb mission, and, to a smaller extent, by the Copernicus Atmosphere Monitoring Service, implemented by the European Centre for Medium-Range Weather Forecasts (ECMWF) on behalf of the European Commission.
OCO-2 L1 and L2 data were produced by the OCO-2 project at the Jet Propulsion Laboratory, California Institute of Technology, and obtained from the ACOS/OCO-2 data archive maintained at the NASA Goddard Earth Science Data and Information Services Center. TCCON data were obtained from the TCCON Data Archive, hosted by the Carbon Dioxide Information Analysis Center (CDIAC) - tccon.onrl.gov. We warmly thank those who made these data available.

## Code/Data availability

The codes used in this paper and the CAMS model simulations are available, upon request, from the author. The OCO-2 and TCCON data can be downloaded from the respective websites.

## Author contributions

FMB designed the study. PC and LD developed the codes and performed the computations. All
authors shared the result analysis.

## Competing interests

The authors declare no competing interests.

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

**Figures and Tables**

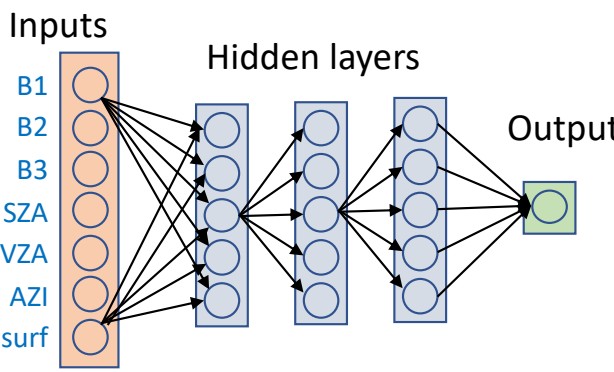


**Figure 1:** Graphical representation of the NN used in this paper. The outputs from all neurons feed in all neurons of the
next layer. There is a variable number of hidden layers. Similarly, there is a choice on the number of
neurons in each layer. Not all inputs are used for the various versions of the NN that are described in this
paper.


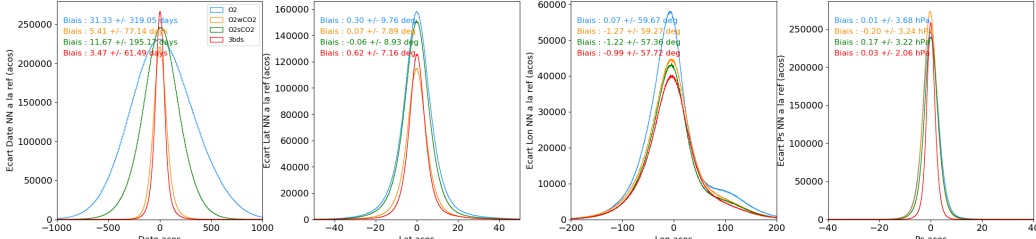

**Figure 2:** Analysis of the ability of the NN to estimate the date, location (latitude, longitude) and surface pressure from
the input spectra and observation geometry. The graphs show the histograms of the differences between
710        the NN estimate and the true value. Several versions of the NN were analyzed using either all three bands
(red), only the wCO2 and O2 bands (orange), sCO2 and O2 (green) and only the O2 band (blue).

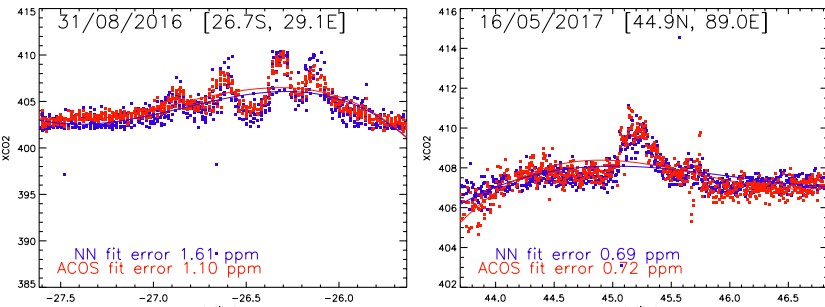

**Figure 3:** Two examples of XCO2 plumes that are captured by the ACOS bias-corrected XCO2 estimates. These were
not shown by the first version of the NN algorithm (shown in Figure A1), but are well captured by the
second version that does not use the wCO2 band (shown here). The NN estimates are in blue whereas the
ACOS estimates are in red. The lines are simple polynomial fits on the XCO2 estimates and do not aim at
capturing the plume signature.



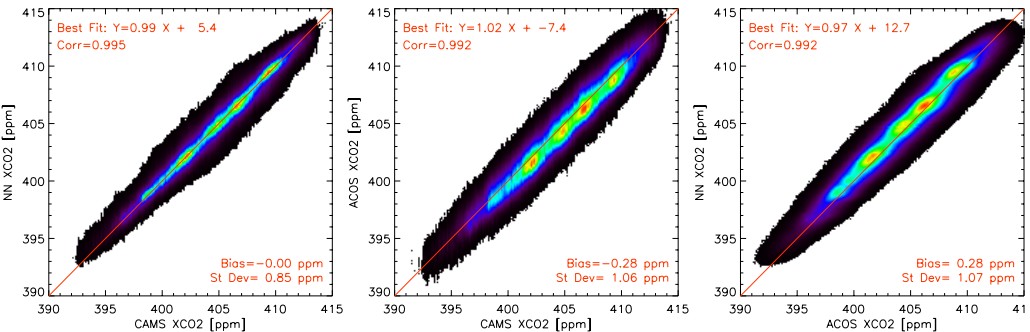


**Figure 4:** Inter-comparison of XCO2 estimated from CAMS, ACOS, and the NN. The density histogram is based on nadir observations from February 2015 do December 2019. A similar figure for the glint cases is shown in the supplementary figure A3.


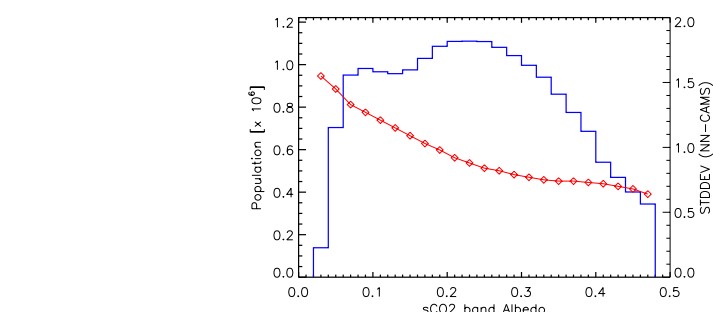

**Figure 5:** Standard deviation of the NN-CAMS difference as a function of the sCO2 band albedo (red, right scale). The computation is made over 0.02 bins, the population of which is shown by the blue line (left scale).


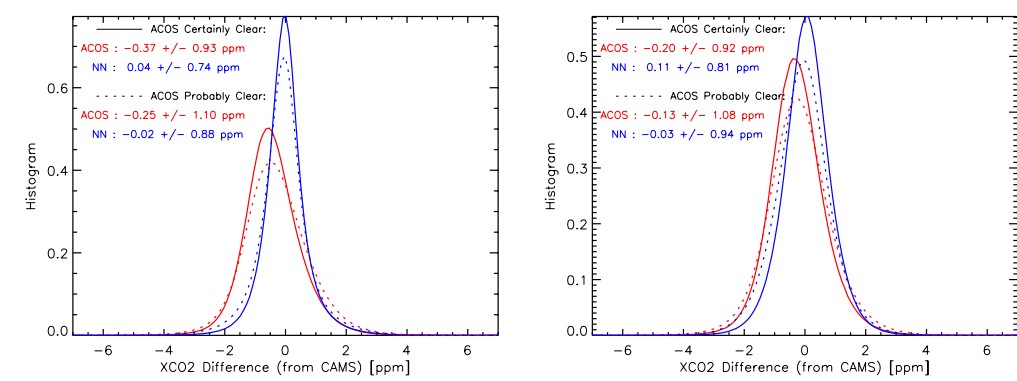

**Figure 6:** Histogram of the differences between either one of the two satellite datasets and the CAMS model. We distinguish cases when the flag *cloud_flag_idp* is "certainly clear" and "probably clear". The left figure is for the nadir dataset, whereas the right figure is for the glint.






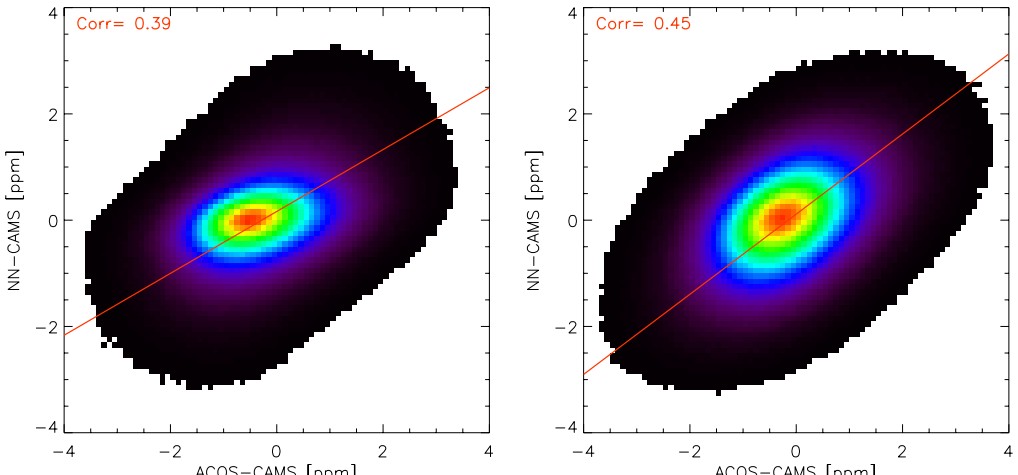

**Figure 7:** Density histogram of the innovation, i.e. the difference between the satellite product and the model estimates. differences between either one of the two satellite datasets and the CAMS model. The red line shows the result of a linear fit through the datapoints aiming at a minimization of the distance to the best line. The left figure is for the nadir dataset, whereas the right figure is for the glint.

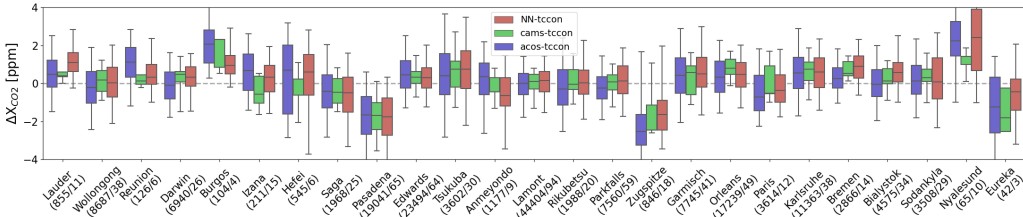

**Figure 8**: Statistics of the differences between the NN retrieval (red), the CAMS model (green) or the bias-corrected ACOS retrievals (blue) and the TCCON retrievals. The boxes indicate the 25-75% percentiles and the median is shown by the horizontal line within the box. The whiskers indicate the 5-95% percentiles. Stations are ordered by increasing latitudes. The numbers below the station name indicate the number of individual observations and coincidence days used for the statistics. The references of the various TCCON observations are provided in table 1. Figure A4 provide similar results for the Glint case





**Table 1**: TCCON stations used in this paper (Figures 8 and A4). The data have been obtained from the tccondata.org web site on Feb 4th, 2021.

| Stations | [ lat ; lon ] | Alt [m] | Reference | Nadir Norb/Nobs | Nadir CAMS / ACOS / NN Biases | Nadir Std Dev | Glint Norb/Nobs | Glint CAMS / ACOS / NN Biases | Glint Std Dev |
|---|---|---|---|---|---|---|---|---|---|
| Lauder | [-45.04 ; 169.68] | 370 | Sherlock et al. 2017 | 11 855 | 0.4 0.5 1.1 | 0.2 1.2 1.0 | 53 5508 | 0.4 0.5 0.8 | 0.3 1.2 1.2 |
| Wollongong | [-34.41 ; 150.88] | 30 | Griffith et al. 2017b | 38 8687 | 0.1 -0.2 0.2 | 0.7 1.3 1.3 | 23 4059 | -0.6 -0.9 -0.7 | 1.2 1.3 1.3 |
| Réunion Island | [-20.90 ; 55.49] | 90 | De Maziere et al. 2017 | 6 126 | 0.2 1.0 0.6 | 0.3 1.4 1.2 | 6 123 | -0.8 -0.5 -1.2 | 0.6 1.4 1.7 |
| Darwin | [-12.43 ; 130.89] | 30 | Griffith et al. 2017a | 26 6940 | 0.4 -0.1 0.4 | 0.7 1.1 1.0 | 28 7432 | 0.4 -0.2 0.4 | 0.6 1.2 1.1 |
| Burgos | [18.53 ; 120.65] | 35 | Morino et al. 2018 | 4 104 | 1.3 2.0 1.5 | 0.7 1.3 1.0 | 9 789 | 0.3 0.6 0.2 | 0.4 1.3 1.1 |
| Izana | [28.3 ; -16.48] | 2300 | Blumenstock et al. 2017 | 15 211 | -0.5 0.6 0.6 | 0.8 1.3 1.1 | 6 183 | -0.1 1.0 0.5 | 0.6 1.4 0.9 |
| Hefei | [31.90 ; 118.67] | 30 | Liu et al. 2018 | 6 545 | -0.2 0.3 0.3 | 1.0 2.1 2.0 | 9 1581 | 0.3 -0.5 -0.1 | 1.1 1.5 1.3 |
| Saga | [33.24 ; 130.29] | 10 | Shiomi et al. 2017 | 25 1968 | -0.4 -0.4 -0.3 | 0.8 1.5 1.6 | 33 2934 | -0.3 -0.5 -0.8 | 0.6 1.4 1.4 |
| Pasadena | [34.14 ; -118.13] | 240 | Wennberg et al. 2017b | 65 19041 | -1.8 -1.7 -1.6 | 1.1 1.5 1.5 | 51 13589 | -1.6 -1.2 -1.5 | 1.1 1.3 1.4 |
| Edwards | [34.96 ; -117.88] | 700 | Iraci et al. 2017 | 64 23494 | 0.3 0.5 0.5 | 0.6 1.2 1.0 | 70 25890 | 0.3 0.6 0.1 | 0.7 1.3 1.1 |
| Tsukuba | [36.05 ; 140.12] | 30 | Morino et al. 2017a | 30 3602 | 0.6 0.5 0.9 | 1.1 1.9 1.7 | 24 1535 | -0.1 -0.1 -0.2 | 0.9 1.5 1.4 |
| Anmeyondo | [36.54 ; 126.33] | 30 | Goo et al. 2014 | 9 1177 | -0.2 0.1 -0.6 | 0.7 1.4 1.5 | 1 396 | 0.2 -0.2 -0.3 | 0.2 1.1 0.5 |
| Lamont | [36.6 ; -97.49] | 320 | Wennberg et al. 2017c | 94 44404 | -0.0 -0.1 0.3 | 0.9 1.0 1.0 | 96 41423 | 0.1 -0.0 0.0 | 0.6 1.0 1.0 |
| Rikubetsu | [43.46 ; 143.77] | 390 | Morino et al. 2017b | 20 1988 | 0.1 -0.2 0.2 | 0.9 1.6 1.3 | 17 2192 | 0.5 0.3 0.3 | 0.5 1.3 1.7 |
| Park Falls | [45.94 ; -90.27] | 440 | Wennberg et al. 2017a | 59 7560 | -0.0 -0.2 0.3 | 0.8 1.0 1.2 | 45 5150 | -0.3 -0.3 -0.4 | 1.0 1.2 1.2 |
| Zugspitze | [47.42 ; 11.06] | 2960 | Sussmann and Rettinger 2017b | 18 846 | -2.0 -2.3 -1.4 | 1.1 1.8 1.6 | 9 1404 | -1.0 -0.2 -0.2 | 1.4 1.6 1.5 |
| Garmisch | [47.48 ; 11.06] | 740 | Sussmann and Rettinger 2017a | 41 7745 | 0.3 0.4 0.8 | 0.9 1.5 1.5 | 38 7248 | 0.3 0.7 0.2 | 0.9 1.4 1.3 |
| Orléans | [47.97 ; 2.11] | 130 | Warneke et al. 2017 | 49 17239 | 0.8 0.3 0.6 | 0.5 1.2 1.1 | 34 10418 | 0.8 0.6 0.5 | 0.7 1.0 1.0 |
| Paris | [48.85 ; 2.36] | 60 | Te et al. 2017 | 12 3614 | -0.0 -0.5 -0.0 | 0.9 1.3 1.2 | 14 3421 | -0.6 -0.4 -0.9 | 1.6 1.7 1.6 |
| Karlsruhe | [49.1 ; 8.44] | 110 | Hase et al. 2017 | 38 11363 | 0.6 0.6 0.8 | 0.7 1.4 1.3 | 45 8883 | 0.4 0.6 0.1 | 0.6 1.2 1.1 |
| Bremen | [53.10 ; 8.85] | 7 | Notholt et al. 2017 | 14 2866 | 0.7 0.3 0.9 | 0.4 0.9 1.1 | 9 1415 | 1.1 0.9 0.6 | 1.0 1.2 1.0 |
| Bialystok | [53.23 ; 23.02] | 180 | Deutscher et al. 2017 | 34 4575 | 0.3 -0.0 0.8 | 0.7 1.3 1.1 | 31 8326 | 0.4 0.2 0.4 | 0.3 1.0 1.1 |
| Sodankylä | [67.37 ; 26.63] | 190 | Kivi et al. 2017 | 29 3508 | 0.4 0.2 0.3 | 0.8 1.3 1.5 | 21 4698 | 0.8 0.8 0.3 | 1.2 1.3 1.7 |
| Ny-Ålesund | [78.9 ; 11.9] | 20 | Notholt et al. 2019 | 10 65 | 1.2 2.1 2.2 | 0.6 2.5 2.3 | 7 18 | 1.0 3.1 1.4 | 0.7 2.9 1.5 |
| Eureka | [80.05 ; -86.42] | 600 | Strong et al. 2017 | 3 42 | -1.4 -1.3 -0.5 | 1.0 2.0 1.9 | 9 429 | 0.6 -0.1 -0.8 | 1.5 2.6 1.7 |



**Appendix**




**Figure A1:** XCO2 estimated by the ACOS algorithm (Red) and the NN approach (Blue) using its
initial version as published in David et al. (2021), as a function of latitude. The ACOS product
shows a number of XCO2 enhancements that are not shown by the NN estimates. The plumes are
observed downwind of large coal power plants, which make these features trustworthy. The date is
August 31st, 2016.

**Figure A2:** Mean difference, at daily scale, between the NN XCO2 estimate and the CAMS model.
The blue dots show the results for the nominal training. The orange dots show the results when the
training was made with a dataset biased by 1 ppm but only the observations of June 2017, the center
of which is indicated by the red vertical line.


**Figure A3:** Same as Figure 4, but for the glint observations.





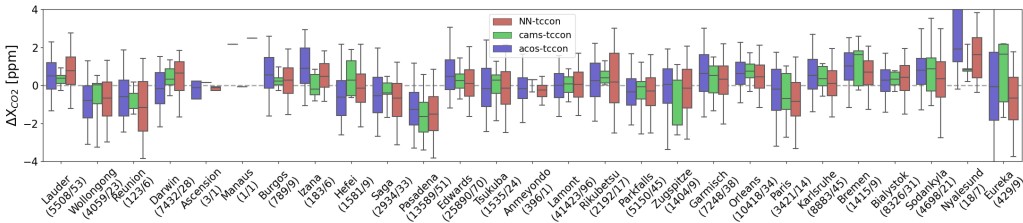


**Figure A4:** Same as Figure 8, but for the glint observations.