# Peer review of "On the potential of a neural network-based approach for estimating XCO2 from OCO-2 measurements"

_Atmospheric Measurement Techniques, 2021_

## Referee Comment (RC1)

Review of "On the potential of a neural network-based approach for estimating XCO2 from OCO-2 measurements" by Breon, David, Chatelanaz, and Chevallier.

**Overview**
This paper follows on from a previous work, David et al. (2021, AMT).  That paper used a neural network approach (NN) to retrieve column mean carbon dioxide concentrations (XCO2) from the Orbiting Carbon Observatory-2 (OCO-2) satellite, directly from the hyperspectral near-infrared radiances recorded by the satellite.  This method is very much faster, and could possibly have smaller biases, than traditional Bayesian "full-physics" retrievals, which in particular require costly, multiple-scattering radiative transfer calculations to be performed for each retrieval.  This new work shows that the previous NN could actually estimate the latitude and date of each observation with surprising accuracy, even though direct information about those quantities was not provided to the NN.  Therefore, it implies that the previous NN was of little value.  This new work in particular showed that plume features could not be recovered from the first NN, a necessary (but not sufficient) step to demonstate that the NN "works".  The authors find that by removing the middle of OCO-2's three spectral bands, the NN can no longer easily reproduce the date (but still can reproduce the latitude with surprising accuracy), and can now reproduce plume features.  This suggests the new NN is indeed properly inferring CO2 from the depth of the spectral absorption features contained within the OCO-2 radiances. The new NN is shown to compare equally well as a standard full physics approach (the ACOS retrieval) to ground-truth measurements from the TCCON network, and has slightly better precision than ACOS as well.

**General Comments**

The findings of the deficiencies of the previous NN (described in David et al., 2021, hereafter D21) are very illuminating and a welcome addition to the literature.  In addition, in this new work they use all 8 OCO-2 cross-track footprints, instead of just a single one.  In fact, they imply (but do not directly ever state) that they use a single NN for all 8 footprints, which would be a significant achievement.  The accuracy and precision of the new NN against TCCON is impressive, as is the finding that the new NN seems to accurately identify and characterize plume features for local fossil fuel sources, such as power plants or urban areas.  However, this paper has a number of shortcomings which must be addressed prior to publication.

Most importantly, like its predecessor paper D21, there are some hypotheses in this paper that are presented as solid truths but in fact may not be so.  In my review of D21, I strongly argued that the presentation by the authors that the NN had learned how to independently estimate XCO2 from the spectra was simply a hypothesis, and suggested that they check well-known plume features (such as from large, isolated power plants) before publication.  They did not do so, and argued against me.  They were proven incorrect, and thankfully state that clearly in this publication.  However, that does not mean that all the stated hypotheses in this new paper, again presented as truths, are indeed so. Primarily, that *now* the NN retrieval with the weak CO2 band removed really does accurately retrieve XCO2 in the way the authors think it does.  Again, this is merely a hypothesis.  Granted, it is supported by the fact that the model can reproduce plume features, but it is by no means proved.  I believe it is certain that the NN is indeed taking some

information from the spectra features directly related to CO2 concentration. However, this does *not* preclude the possibility that there still may be other features in the spectra that the NN could be using to reproduce features of the CAMS model, features themselves which indeed may be incorrect in reality.

For instance, the CAMS model differs from other models in certain areas of the globe. It is possible that CAMS is more accurate than the others, but the reverse is also possible. Therefore, why risk using the CAMS model to train the NN over times & places where it significantly disagrees with other models? We are explicitly trying to figure out which model is more correct by using satellite data. If the NN is somehow replicating CAMS biases, we would have no easy way of knowing. And therefore, the NN results will always be suspect. Others (like me!) may suspect that the NN-derived OCO-2 values agree better with CAMS simply because it was trained on CAMS, not because CAMS is actually correct. I understand that the authors' goals are merely to show the potential of a NN approach. But this is also linked, I believe, to extremely careful training. If we have learned anything from D21, it is that the NN can learn ways to predict things in very different ways than you think it does. It tricked the authors in D21. It can do so again, unless the authors are extraordinarily careful and do many supporting checks to ensure that this is not the case. I'm not convinced that this is sufficiently done in the current manuscript.

One simple test is to retrain the data on relatively uninteresting soundings in places where there is not a lot of disagreement among models. For instance, OCO-2 results over the Amazon region and the Sahel region of Africa, as well as eastern China, are all areas of some disagreement and debate (see for instance Peiro et al., 2021, ACP, https://acp.copernicus.org/preprints/acp-2021-373/). Why not remove these areas from the training? Surely the rest of the globe has adequate ranges of surface albedo, viewing geometry, aerosols, etc., that XCO2 in other regions should be sufficient to teach the NN how to retrieve XCO2 in these regions? I would like to see more tests like this to strengthen the findings. If the authors insist that such tests are "beyond the scope of this work", then statements about success of this NN *must* be toned, or given appropriate caveats, prior to publication.

The other main critique is that the authors heavily rely on ACOS quality filtering to select soundings on which to retrieve. This is a major difficulty faced by all satellite XCO2 retrievals. ACOS uses a number of variables on which to screen data for retrieval, as described in detail by O'Dell et al. (2018, AMT). It is not at all clear how the NN could address this. The authors suggest that by using the difference from the NN-retrieved to the prior surface pressure, it would be "easy" to accomplish that goal. But they do not show this to be the case, and in my experience many other variables besides that one will come into play. Without other variables to help indicate quality (such as goodness of fit statistics, albedo mean and slope retrieval discrepancies, retrieved aerosol, etc), it is unclear if it is indeed possible at all to accomplish this with a standalone NN. This should be stated clearly in the discussion section, that this is an unsolved problem.

Beyond these critiques, there are additional questions & suggestions given below which must be adequately addressed prior to publication.

**Specific Comments**

L83: "The uncertainties

L79: "Our hypothesis was that the CAMS … model constrained by surface air-sample measurements provides a fairly accurate estimate of the atmospheric $CO_2$ concentration, including the growth rate over multiple years." Please provide evidence for this statement. (e.g., https://atmosphere.copernicus.eu/sites/default/files/custom-uploads/EQC-GHG/CAMS73_2018SC2_D73.1.4.1-2020-v5_202109_v1.pdf, Chevallier et al., 2019, https://acp.copernicus.org/articles/19/14233/2019/).

L83: "The uncertainties on the modeling are small with respect to the range of XCO2 samples that is available in the multi-year dataset." Please defend this statement quantitatively. How big are each?

L105: Was a single NN used for all 8 footprints, or did you train 8 different NN's? Please state clearly in the main text. It's relevant, because the line features move around due to the slightly different wavelength calibration of each footprint. Ie, channel 500 of footprint 1 is not at the same wavelength as channel 500 of the other footprints.

Section3 / Figure A1: Since one of the main points of this paper is to discuss the failure of the first NN and how it was improved, showing the failure in the main text is critical. Therefore, the failure of the first NN to find plumes should be figure 1 rather than A1. Also, because there can be "false plumes" in the OCO-2 data associated with dust or other aerosol features, it is important that you know that the plume seen by ACOS is *real*. How do we know that the multiple plumes in fig A1 are not some source of ACOS-induced bias? Therefore, this figure requires you to use a documented case caused by a known urban or power plant emission source. Many examples abound, for example Nassar et al. (2021, *RSE*, https://doi.org/10.1016/j.rse.2021.112579) and Reuter et al. (2019, *ACP*, https://doi.org/10.5194/acp-19-9371-2019).

Figure 3: Please list the fossil fuel sources of the plumes. If you cannot, please use other examples where the source is documented, again so that we know that these plumes are real (see previous comment). If possible, cite supporting sources.

L184: "Standard deviation of the latitude estimate". I think the authors mean "Standard devation of the latitude error". Please correct. Similarly for the statements about the longitude and date errors.

L183-215: Regarding the estimate of date & latitude. Can you please state whether the accuracy on these variables was independent of footprint or not? Ie, was it different for footprints 1-8 at all? Often, calibration artifacts such as bad pixels affect the different footprints a little differently, so if it is dependent on footprint, that would tell you if it was more likely to be some calibration artifact that the NN is keying off of for its estimates.

L223: Please repeat this analysis for the O2+sCO2 NN results (sigma_lat = 8.9 deg, sigma_lon=57 deg, sigma_date = 195 days), and state the resulting XCO2 accuracy, to show that the inherent accuracy from latitude and date alone is relatively poor for that band combination, further justifying the second version of the NN.

L248: You may also wish to state that the use of the NWP surface pressure as input to your NN is further justified considering the fact that the ACOS algorithm also explicitly uses it in its posterior bias correction, and in fact it is the most important term in the bias correction (O'Dell et al., 2018).

L262: "there is no satellite data input to CAMS". The informed reader will know that this is not true for all versions of CAMS. FT20r3, for example, assimilates OCO2 rather than surface/in-situ data. As you report the standard deviation of your result vs. CAMS (0.85 ppm), it may also be interesting to report the same but for the CAMS version which assimilates OCO2. If your hypothesis is true, that standard deviation should be lower.

L291: Please define and justify the statement "significantly correlated". The R-value for land nadir is merely 0.39 as shown in your figure 7; which seesm to imply that only 15% of the innovation difference variance is common to the two datasets. Some of this may be due to instrument noise, which you could reduce by averaging up the data (say to all soundings that fall in a given 10-second block, as is commonly done by modelers, see for example Peiro et al., 2021, ACP, https://acp.copernicus.org/preprints/acp-2021-373/). Further, the best-fit line appears to fall significantly away from the 1-1 line. However, that could be to due more "noise" in the ACOS fit.

L322: I think you mean that the comparison to TCCON does not *suggest* favoring one satellite product of the other. It would allow it if there were any obvious difference, it just doesn't suggest it with this analysis.

L326: Your statement on the value of the satellite data relative to the CAMS model makes little sense. There are many models in addition to CAMS, and they disagree about many, many things of importance to the carbon cycle. The TCCON data seem to have limited value in resolving most of these questions, especially in the tropics where the TCCON data are incredibly sparse. In addition, the in-situ-driven CAMS results typically run 12 months behind real-time, while satellite data are available within 1 month of data collection. Indeed, this was the motivation behind the CAMS "FastTrack" (FT) product, which assimilates OCO-2 rather than in-situ data, and has been shown to compare equally well with independent aircraft data (Chevallier et al., 2019, ACP, https://doi.org/10.5194/acp-19-14233-2019). Please modify or remove this statement.

L335: Regarding your statement on the "agreement with CAMS": You seem to imply that the better agreement with CAMS for the NN implies that the NN product is "better" than ACOS. You simply cannot draw this conclusion when the NN was trained to agree with CAMS. If it didn't agree better with CAMS than ACOS, something would be wrong. The agreement with CAMS tells you literally nothing about the quality of the NN beyond the fact that it has been properly trained. Please rephrase this statement to reflect this fact.

Discussion section: Please also mention / highlight the fact that (if I've interpreted your paper correctly), the same NN training was applied to all 8 OCO-2 footprints. That's quite amazing. If so, it's necessary to perform a brief analysis on the quality of the XCO2 analysis from the 8 different footprints. Are they all comparable? If so, this is a remarkable result given that the NN does not "know" a-priori the wavelength grid of each footprint. If not, it is important to know if each footprint is required to be treated with a separate NN training. This is important for future sensors such as CO2M and GeoCarb, which may have 100s to 1000 different cross-track footprints (and thus training 1000 different NN's may be challenging).

L400: Using the surface pressure difference to the met forecast *might* provide such a quality flag. It might not. It's a hypothesis that would need to be tested. ACOS uses many variables, both pre- and post-retrieval, as indicators of quality, of which surface pressure error is just one.

**Technical Comments**

L155: as input, the training → as input, and the training
L157: as → in that
L159: worrisome however. → worrisome, however.
L160: well documented → well-documented
L160-1: local enhancement → local enhancements;  plume → plumes
L162 : South African → South Africa
L195 : a combination of O2 band with either CO2 bands → a combination of the O2 band with either CO2 band
L212: provides an indirect information → provides indirect information
L240: shown on Figure 3 → shown in Figure 3
L253: leads to a slightly better → leads to slightly better
L295: remotely sensed → remotely-sensed
L321: agreements are → agreement is
L344: contrarily → contrary
L365: provides → provided (to keep with the same verb tense as this earlier finding provided motivation for the present study)

---

## Author Comment (AC1)

In the following, the review is shown in plain text, our answers are in blue, and text that has been added to the manuscript is in green.

**Reviewer 1 : Chris O'Dell**

We warmly thank Chris O'Dell for the time he spent on his in-depth review. We also acknowledge the fact that, for our first paper that he also commented, he did interpret that our NN approach was mostly reproducing the training dataset while we argue against it. He was right, and we were wrong.

Review of "On the potential of a neural network-based approach for estimating XCO2 from OCO-2 measurements" by Breon, David, Chatelanaz, and Chevallier.

**Overview**
This paper follows on from a previous work, David et al. (2021, AMT). That paper used a neural network approach (NN) to retrieve column mean carbon dioxide concentrations (XCO2) from the Orbiting Carbon Observatory-2 (OCO-2) satellite, directly from the hyperspectral nearinfrared radiances recorded by the satellite. This method is very much faster, and could possibly have smaller biases, than traditional Bayesian "full-physics" retrievals, which in particular require costly, multiple-scattering radiative transfer calculations to be performed for each retrieval. This new work shows that the previous NN could actually estimate the latitude and date of each observation with surprising accuracy, even though direct information about those quantities was not provided to the NN. Therefore, it implies that the previous NN was of little value. This new work in particular showed that plume features could not be recovered from the first NN, a necessary (but not sufficient) step to demonstrate that the NN "works". The authors find that by removing the middle of OCO-2's three spectral bands, the NN can no longer easily reproduce the date (but still can reproduce the latitude with surprising accuracy), and can now reproduce plume features. This suggests the new NN is indeed properly inferring CO2 from the depth of the spectral absorption features contained within the OCO-2 radiances. The new NN is shown to compare equally well as a standard full physics approach (the ACOS retrieval) to ground-truth measurements from the TCCON network, and has slightly better precision than ACOS as well.
This is a good summary of our work, although we may argue below on the "but not sufficient"

**General Comments**
The findings of the deficiencies of the previous NN (described in David et al., 2021, hereafter D21) are very illuminating and a welcome addition to the literature. In addition, in this new work they use all 8 OCO-2 cross-track footprints, instead of just a single one. In fact, they imply (but do not directly ever state) that they use a single NN for all 8 footprints, which would be a significant achievement. The accuracy and precision of the new NN against TCCON is impressive, as is the finding that the new NN seems to accurately identify and characterize plume features for local fossil fuel sources, such as power plants or urban areas. However, this paper has a number of shortcomings which must be addressed prior to publication.
Again, this is a good summary, and we appreciate the "welcome addition to the literature"

Most importantly, like its predecessor paper D21, there are some hypotheses in this paper that are presented as solid truths but in fact may not be so. In my review of D21, I strongly argued that the presentation by the authors that the NN had learned how to independently estimate XCO2 from the spectra was simply a hypothesis, and suggested that they check well-known plume features (such as from large, isolated power plants) before publication. They did not do so, and argued against me. They were proven incorrect, and thankfully state that clearly in this publication.
We agree. The technical maturity of our work at the time did not allow us to look at plume easily because, as we explained at the time, we were limited to a single cross-track FOV within the eight of the instrument, which hampered the identification of plumes along the orbit track. This limitation came in addition to known false plume cases in the retrievals, acknowledged by the review later in his text. However, Chris O'Dell was right, and we made an erroneous interpretation of our results, believing it

was "impossible" for the NN to know about the observation date (in particular the year of observation). We believe we do better this time.

However, that does not mean that all the stated hypotheses in this new paper, again presented as truths, are indeed so. Primarily, that now the NN retrieval with the weak CO2 band removed really does accurately retrieve XCO2 in the way the authors think it does. Again, this is merely a hypothesis.

The hypothesis is based on
- Agreement with the CAMS model
- Agreement with TCCON
- Innovation (difference with the model) that is correlated with ACOS product
We feel these are rather strong arguments to support the hypothesis

Granted, it is supported by the fact that the model can reproduce plume features, but it is by no means proved. I believe it is certain that the NN is indeed taking some information from the spectra features directly related to CO2 concentration. However, this does not preclude the possibility that there still may be other features in the spectra that the NN could be using to reproduce features of the CAMS model, features themselves which indeed may be incorrect in reality.
There are two arguments that indicate the NN is not only based on the CAMS training
- The depiction of plumes that are not described by CAMS
- The innovation (difference with CAMS) than somewhat agrees with ACOS (Figure 7)
But yes, the NN estimate is not independent from CAMS and how much comes from CAMS or the spectra is not evaluated.

For instance, the CAMS model differs from other models in certain areas of the globe. It is possible that CAMS is more accurate than the others, but the reverse is also possible. Therefore, why risk using the CAMS model to train the NN over times & places where it significantly disagrees with other models?
We do not claim that CAMS is the best model.    We used it for practical purposes (as one of the co-authors is the main developer).    The comparison against TCCON indicates that CAMS is "good" and the NN results indicate it is "good enough" for our purposes.    But yes, another model may lead to even better results.    It would be an interesting exercise to train the NN with different model and analyze (i) the ability of the NN to reproduce the model simulations and (ii) the differences between the various NN prediction.
We have added a few sentences to explain the choice of CAMS : Note that other 4D description of the atmospheric composition could have been used for our work.    We chose CAMS mostly for practical reasons and the same procedure may be attempted with another modeling dataset.

We are explicitly trying to figure out which model is more correct by using satellite data. If the NN is somehow replicating CAMS biases, we would have no easy way of knowing. And therefore, the NN results will always be suspect. Others (like me!) may suspect that the NN-derived OCO-2 values agree better with CAMS simply because it was trained on CAMS, not because CAMS is actually correct.
We agree with the reviewer concern that the NN may only reproduce the CAMS description of the atmosphere, so that the satellite product, whose main objective is to improve our current knowledge that is implicitly described in the 4D modeling, would be useless. This is why the NN product evaluation must be achieved not only against CAMS, but also against independent information. The TCCON observation is an independent set of data but the comparison does not demonstrate that the satellite product (neither the NN not ACOS) does any better that the CAMS product.    We have added the sentences Let us stress that any bias in CAMS may be transferred to the NN product.    Thus, a high agreement between CAMS and the NN product is not a demonstration of the latter accuracy.

I understand that the authors' goals are merely to show the potential of a NN approach. But this is also linked, I believe, to extremely careful training. If we have learned anything from D21, it is that the NN can learn ways to predict things in very different ways than you think it does. It tricked the authors in D21. It can do so again, unless the authors are extraordinarily careful and do many supporting checks to ensure that this is not the case. I'm not convinced that this is sufficiently done in the current

manuscript.

We definitely agree with the reviewer call for caution. We have been tricked once, and may very well be again. We have added these sentences as a final remark to the manuscript

As a final remark, we call for caution. We have been tricked by the NN ability to generate a consistent description of the atmospheric XCO2 in our first analysis. It is difficult to ensure that we are not tricked again. The source of the information that leads to a fairly accurate estimate of the date, when using the weak CO2 band, remains unclear. As a consequence, although it is demonstrated that the new version of the NN generates structures that are not in the training dataset, there may be biases in the CAMS modeling that have a significant influence on the NN product.

One simple test is to retrain the data on relatively uninteresting soundings in places where there is not a lot of disagreement among models. For instance, OCO-2 results over the Amazon region and the Sahel region of Africa, as well as eastern China, are all areas of some disagreement and debate (see for instance Peiro et al., 2021, ACP, https://acp.copernicus.org/preprints/acp-2021- 373/). Why not remove these areas from the training? Surely the rest of the globe has adequate ranges of surface albedo, viewing geometry, aerosols, etc., that XCO2 in other regions should be sufficient to teach the NN how to retrieve XCO2 in these regions? I would like to see more tests like this to strengthen the findings. If the authors insist that such tests are "beyond the scope of this work", then statements about success of this NN must be toned, or given appropriate caveats, prior to publication.

We agree that the principle of the NN approach that we develop should be resilient to the removal of an entire region for the training. Amazonia is not the optimal region for the suggested experiment because the South Atlantic anomaly leads to a very high fraction of corrupted spectra.

Also, choosing an area where the model may be wrong does not seem the best option to evaluate the NN. Thus we favor the following experiment : Using CAMS, train the NN based on a set of observations distributed over the globe but excluding Europe; and then evaluate the NN estimate over Europe against the CAMS model. We shall do that and other experiment using different modeling in a subsequent analysis.

The other main critique is that the authors heavily rely on ACOS quality filtering to select soundings on which to retrieve. This is a major difficulty faced by all satellite XCO2 retrievals. ACOS uses a number of variables on which to screen data for retrieval, as described in detail by O'Dell et al. (2018, AMT). It is not at all clear how the NN could address this. The authors suggest that by using the difference from the NN-retrieved to the prior surface pressure, it would be "easy" to accomplish that goal. But they do not show this to be the case, and in my experience many other variables besides that one will come into play. Without other variables to help indicate quality (such as goodness of fit statistics, albedo mean and slope retrieval discrepancies, retrieved aerosol, etc), it is unclear if it is indeed possible at all to accomplish this with a standalone NN. This should be stated clearly in the discussion section, that this is an unsolved problem.

We certainly agree that the NN approach that is described in the manuscript relies on the ACOS cloud screening, so that our processing chain is NOT independent from ACOS. We acknowledged this fact in the discussion and conclusion section, but we now make it even clearer. We are currently working on the cloud detection based on the NN approach and it will be described in another article.

We have extended the discussion section to make that clear

The first challenge is the cloud detection. All the analysis described in this paper relies on the ACOS cloud detection and only the observations identified as "clear" are processed. Our analysis demonstrates the potential of the NN approach but is currently not independent from ACOS. We are currently evaluating independent approaches for the cloud detection. Although the NN described here aims at an estimate of XCO2, we…

Beyond these critiques, there are additional questions & suggestions given below which must be adequately addressed prior to publication.

**Specific Comments**

L79: "Our hypothesis was that the CAMS … model constrained by surface air-sample measurements provides a fairly accurate estimate of the atmospheric $CO_2$ concentration, including the growth rate over multiple years." Please provide evidence for this statement. (e.g., https://atmosphere.copernicus.eu/sites/default/files/custom-uploads/EQCGHG/ CAMS73_2018SC2_D73.1.4.1-2020-v5_202109_v1.pdf, Chevallier et al., 2019)
We now make reference to the suggested publication and also to Figure 8

L83: "The uncertainties on the modeling are small with respect to the range of XCO2 samples that is available in the multi-year dataset." Please defend this statement quantitatively. How big are each?
We have added this information in the text. From Figure 4 center, one may deduce a standard deviation of 1 ppm or less and 20 ppm.

L105: Was a single NN used for all 8 footprints, or did you train 8 different NN's? Please state clearly in the main text. It's relevant, because the line features move around due to the slightly different wavelength calibration of each footprint. Ie, channel 500 of footprint 1 is not at the same wavelength as channel 500 of the other footprints.
We use a single NN for all footprint. We make that clear in the revised version. The NN may deduce information about the footprint from the VZA, the azimuth and the other pieces of information that are provided as input.

Section3 / Figure A1: Since one of the main points of this paper is to discuss the failure of the first NN and how it was improved, showing the failure in the main text is critical. Therefore, the failure of the first NN to find plumes should be figure 1 rather than A1.
In our opinion, a figure must be used if it brings a message further than what can be described in the text. Here, the message is "there is no plume feature in the NN XCO2 estimate" and the figure does not convey further information so that, in our opinion, it is not useful for the main body of the paper.

Also, because there can be "false plumes" in the OCO-2 data associated with dust or other aerosol features, it is important that you know that the plume seen by ACOS is real. How do we know that the multiple plumes in fig A1 are not some source of ACOS-induced bias? Therefore, this figure requires you to use a documented case caused by a known urban or power plant emission source. Many examples abound, for example Nassar et al. (2021, RSE, https://doi.org/10.1016/j.rse.2021.112579) and Reuter et al. (2019, ACP, https://doi.org/10.5194/acp-19-9371-2019).
We agree with the reviewer comment that it is much better to use cases that have been identified and analyzed in the literature. In the revised version, we have changed the cases that are used in the main body of the manuscript to two cases identified and discussed in in Reuter 2019 and Nassar 2021. In addition, we show in the supplementary the figure (XCO2 as a function of latitude) for several other cases identified and discussed in these two papers.

Figure 3: Please list the fossil fuel sources of the plumes. If you cannot, please use other examples where the source is documented, again so that we know that these plumes are real (see previous comment). If possible, cite supporting sources.
See our response to previous comment.

L184: "Standard deviation of the latitude estimate". I think the authors mean "Standard deviation of the latitude error". Please correct. Similarly for the statements about the longitude and date errors.
Indeed. Thanks for the catch. Corrected.

L183-215: Regarding the estimate of date & latitude. Can you please state whether the accuracy on these variables was independent of footprint or not? Ie, was it different for footprints 1-8 at all? Often, calibration artifacts such as bad pixels affect the different footprints a little differently, so if it is dependent on footprint, that would tell you if it was more likely to be some calibration artifact that the NN is keying off of for its estimates.
There was no significant difference with the footprint. This information is added in the revised version.

L223: Please repeat this analysis for the O2+sCO2 NN results (sigma_lat = 8.9 deg, sigma_lon=57 deg, sigma_date = 195 days), and state the resulting XCO2 accuracy, to show that the inherent accuracy from latitude and date alone is relatively poor for that band combination, further justifying the second version of the NN.

We have done the experiment suggested by the reviewer and added a sentence: .   The important point is that the error increases considerably (a factor of 2) for degraded precisions on the location and date with a different version of the NN that is discussed below.

L248: You may also wish to state that the use of the NWP surface pressure as input to your NN is further justified considering the fact that the ACOS algorithm also explicitly uses it in its posterior bias correction, and in fact it is the most important term in the bias correction (O'Dell et al., 2018).

Thanks, we have complemented the sentence (part in italic): However, the surface pressure may alternatively provide useful information to the NN for the interpretation of the spectra, as it does in the full-physics algorithms in the form of a prior estimate *and also for the derivation of the bias corrected product*.

L262: "there is no satellite data input to CAMS". The informed reader will know that this is not true for all versions of CAMS. FT20r3, for example, assimilates OCO2 rather than surface/insitu data. As you report the standard deviation of your result vs. CAMS (0.85 ppm), it may also be interesting to report the same but for the CAMS version which assimilates OCO2. If your hypothesis is true, that standard deviation should be lower.

We felt it was clear that the sentence refers to the version of CAMS that is used in the paper.   We nevertheless modified the sentence to confirm that we are referring to a different version of CAMS that does not use satellite data as indicated earlier in the paper: let us recall that there is no satellite data input to the version of CAMS that is used here, so that it is fully independent from ACOS.

L291: Please define and justify the statement "significantly correlated". The R-value for land nadir is merely 0.39 as shown in your figure 7; which seems to imply that only 15% of the innovation difference variance is common to the two datasets. Some of this may be due to instrument noise, which you could reduce by averaging up the data (say to all soundings that fall in a given 10-second block, as is commonly done by modelers, see for example Peiro et al., 2021, ACP, https://acp.copernicus.org/preprints/acp-2021-373/). Further, the best-fit line appears to fall significantly away from the 1-1 line. However, that could be to due more "noise" in the ACOS fit.

We feel that the necessary information, in the number and Figure 7, is provided.   The two variables are definitely correlated.   Indeed, the noise limits the correlation but it does show that the NN is not a mere noisy copy of CAMS which is the message of this figure

L322: I think you mean that the comparison to TCCON does not suggest favoring one satellite product of the other. It would allow it if there were any obvious difference, it just doesn't suggest it with this analysis.

We agree. This is clearly said (we feel) in the sentence "The comparison with TCCON does not allow favoring one satellite estimate versus the other."

L326: Your statement on the value of the satellite data relative to the CAMS model makes little sense. There are many models in addition to CAMS, and they disagree about many, many things of importance to the carbon cycle. The TCCON data seem to have limited value in resolving most of these questions, especially in the tropics where the TCCON data are incredibly sparse. In addition, the in-situ-driven CAMS results typically run 12 months behind real-time, while satellite data are available within 1 month of data collection. Indeed, this was the motivation behind the CAMS "FastTrack" (FT) product, which assimilates OCO-2 rather than in-situ data, and has been shown to compare equally well with independent aircraft data (Chevallier et al., 2019, ACP, https://doi.org/10.5194/acp-19-14233-2019). Please modify or remove this statement.

The sentence was modified as follows
The figure (and table) also clearly shows that the CAMS product offers a better agreement with the TCCON data than any of the satellite estimates in most cases. The high quality of the CAMS modelling used in this paper, at least over the TCCON site, provides further justification of its use as a training dataset.

L335: Regarding your statement on the "agreement with CAMS": You seem to imply that the better agreement with CAMS for the NN implies that the NN product is "better" than ACOS. You simply cannot draw this conclusion when the NN was trained to agree with CAMS. If it didn't agree better with CAMS than ACOS, something would be wrong. The agreement with CAMS tells you literally nothing about the quality of the NN beyond the fact that it has been properly trained. Please rephrase this statement to reflect this fact.
Agreed. We rephrased the sentence. The agreement with CAMS is slightly degraded with respect to the nadir cases (0.92 ppm vs 0.85 for the "certainly clear" observations) but somewhat closer than that of ACOS (Fig. 6 and S3). However, this paragraph is only an application to the glint case and a confirmation of the conclusion obtained for Nadir. Thus, the most important is the longer discussion made for the Nadir results.

Discussion section: Please also mention / highlight the fact that (if I've interpreted your paper correctly), the same NN training was applied to all 8 OCO-2 footprints. That's quite amazing. If so, it's necessary to perform a brief analysis on the quality of the XCO2 analysis from the 8 different footprints. Are they all comparable? If so, this is a remarkable result given that the NN does not "know" a-priori the wavelength grid of each footprint. If not, it is important to know if each footprint is required to be treated with a separate NN training. This is important for future sensors such as CO2M and GeoCarb, which may have 100s to 1000 different cross-track footprints (and thus training 1000 different NN's may be challenging).
Following this request, we made an analysis of the satellite-CAMS difference for the various footprints. The difference are rather small except for FP #2 that shows a slightly lower performance for both satellite products (i.e. ≈ 1 ppm versus 0.9 ppm standard deviation). We made it clear in the revised manuscript that a single NN is used for all FP, and we added two sentences on this quick analysis of the NN performance.
Note that we use here a single neural network for the eight footprints of the OCO-2 instrument. We analyzed whether the result performance, assessed as the standard deviation of the differences with CAMS, is a function of the footprint. The statistics are very similar for all, except for footprint #2 that shows slightly higher deviation for both the ACOS and the NN satellite products (a difference of ≈0.1 ppm to the mean of ≈1 ppm).

L400: Using the surface pressure difference to the met forecast *might* provide such a quality flag. It might not. It's a hypothesis that would need to be tested. ACOS uses many variables, both pre- and post-retrieval, as indicators of quality, of which surface pressure error is just one.
We certainly agree that we this hypothesis needs to be tested. Actually, current tests are not performing as well as expected. We have added the sentence "This idea remains to be evaluated."

**Technical Comments**
All technical comments have been accounted for as suggested by the reviewer. We thank him, again, for taking the time to correct our writing.

L155: as input, the training à as input, and the training

L157: as à in that

L159: worrisome however. à worrisome, however.

L160: well documented à well-documented

L160-1: local enhancement à local enhancements; plume à plumes

L162 : South African à South Africa

L195 : a combination of O2 band with either CO2 bands à a combination of the O2 band with either CO2 band

L212: provides an indirect information à provides indirect information

L240: shown on Figure 3 à shown in Figure 3

L253: leads to a slightly better à leads to slightly better

L295: remotely sensed à remotely-sensed

L321: agreements are à agreement is

L344: contrarily à contrary

L365: provides à provided (to keep with the same verb tense as this earlier finding provided motivation for the present study)

---

## Author Comment (AC2)

In the following, the review is shown in plain text, our answers are in blue, and text that has been added to the manuscript is in green.

**Reviewer 2 : Sihe Chen**

**Overview**
This paper corrects a problem in a previous work, David et al., (2021). In that work, the NN is overfitted to be able to predict the latitude of the sounding. In this work, the problem from the previous work is fixed, and the weak CO2 band is excluded from the analysis. Good results are obtained in comparison to different XCO2 sources.

**Comments:**
The article provides a very good general network trained for prediction of XCO2. I recommend that this article be added to the literature with several of the following minor issues addressed.
We thank the reviewer for his work on our paper and for the positive comments

The authors have clearly shown a nice hypothesis of how D21 could have made a precise prediction on its location. From my perspective of view, this hypothesis is not hard to test. For example, the authors could try to train a NN with wCO2 as the input only and try to retrieve the time information, or they can show a figure just like Figure 1 for an NN with wCO2 removed.
We have made a number of attempts to try to understand the information content that is used by the Neural Network. Our analysis shows that, as explained in the manuscript, the NN behavior is very different when the wCO2 band is used, or not, as input. We do not understand the reviewer's suggestion concerning Figure 1. The reviewer may have meant Figure A1, in which case this is what we have done : Figure A1 shows that no plume feature is retrieved by the NN when the wCO2 band is used. Conversely, this feature is retrieved when it is not used (Figure 3).

Regarding the description of NN: can you provide the loss function that is used? Also, for the NN structure, I suggest that you show what specific hidden layer numbers are chosen in a table and how they are chosen. Doing something similar for the number of hidden layers could be good too, which should be better than simply stating the decision to be related to experiences. An example is Chen et al., 2022, fig. 8: https://www.sciencedirect.com/science/article/pii/S0022407321005409
We use the Mean Absolute Error as the loss function. We have added the information suggested by the reviewer in the revised version of the manuscript

Line 267: At first it was stated that sCO2 band was considered but later an O2 band albedo increase is specified. Also, it would be good if both bands' albedo could be compared to the standard deviation, so that we can see which factor plays a more important role.
Indeed, there was a typo. Only the sCO2 band was supposed to be mentioned. We made the correction (thanks!) and added a sentence: The estimate precision is also a function of the O2 band albedo, but this effect is not as strong and the O2 band albedo shows less variability than that of the sCO2 band.